# Learning from negative feedback,
# or positive feedback or both

**Abbas Abdolmaleki, Bilal Piot, Bobak Shahriari, Jost Tobias Springenberg**
**Tim Hertweck, Rishabh Joshi, Junhyuk Oh, Michael Bloesch, Thomas Lampe**
**Nicolas Heess, Jonas Buchli, Martin Riedmiller**
Google DeepMind *

## Abstract

Existing preference optimization methods often assume scenarios where paired preference feedback (preferred/positive vs. dis-preferred/negative examples) is available. This requirement limits their applicability in scenarios where only unpaired feedback—for example, either positive or negative— is available. To address this, we introduce a novel approach that decouples learning from positive and negative feedback. This decoupling enables control over the influence of each feedback type and, importantly, allows learning even when only one feedback type is present. A key contribution is demonstrating stable learning from negative feedback alone, a capability not well-addressed by current methods. Our approach builds upon the probabilistic framework introduced in (Dayan & Hinton, 1997), which uses expectation-maximization (EM) to directly optimize the probability of positive outcomes (as opposed to classic expected reward maximization). We address a key limitation in current EM-based methods: they solely maximize the likelihood of positive examples, while neglecting negative ones. We show how to extend EM algorithms to explicitly incorporate negative examples, leading to a theoretically grounded algorithm that offers an intuitive and versatile way to learn from both positive and negative feedback. We evaluate our approach for training language models based on human feedback as well as training policies for sequential decision-making problems, where learned value functions are available.

## 1 Introduction

The use of preference annotated data for training machine learning models has a long history going back to early algorithms for recommender systems and market research (Guo & Sanner, 2010; Boutilier, 2002; Bonilla et al., 2010). These days preference optimization algorithms are receiving renewed attention since they are a natural candidate for shaping the outputs of deep learning systems, such as large language models (Ouyang et al., 2022; Team, 2024a) or control policies, via human feedback (Christiano et al., 2017; Rafailov et al., 2023; Azar et al., 2023). Arguably, preference optimization algorithms can also be a natural choice even when direct human feedback is not available but one instead aims to optimize a machine learning model based on feedback from a hand-coded or learned critic function (judging desirability of solutions). Here preference optimization methods are useful since they let us optimize the model to achieve desired outcomes based on relative rankings between outcomes alone (rather than requiring absolute labels or carefully crafted reward functions).

Among preference optimization approaches, those based on directly using preference data – as opposed to casting preference optimization as reinforcement learning from (human) feedback – such as DPO (Rafailov et al., 2023), have emerged as particularly successful since they only require access to an offline dataset of paired preference data, and are fairly robust to application domain and hyperparameter settings. However, algorithms within this class make specific assumptions tailored to their application domain. They were designed to optimize LLMs from human feedback in the form of comparisons of generated sentences and thus, by design, require paired preference data (since they directly model a specific choice of preference distribution). We are interested in finding algorithms that are more flexible, and applicable in settings where the assumptions underlying DPO do not apply.

---

*Corresponding Author: Abbas Abdolmaleki <aabdolmaleki@google.com>

In this work we take a fresh look at preference optimization from a probabilistic inference perspective that has been used with great success in the literature on KL regularized reinforcement learning (Dayan & Hinton, 1997; Peters et al., 2010; Abdolmaleki et al., 2018). We find that from this perspective a simplified approach to preference optimization can be derived that is intuitive to understand and is capable of leveraging an arbitrary number of unpaired preferred or dis-preferred outcomes, or even solely one type (positive or negative) of preference feedback. In particular, our method is able to learn even if exclusively positive or negative examples are available. Formally, our method involves an objective consisting of three log likelihood terms that are derived from first principles: maximizing the likelihood of preferred outcomes, minimizing the likelihood of dis-preferred outcomes, while staying close to a reference distribution (see equation 10). We show the effectiveness of our method across a wide range of benchmarks including synthetic benchmarks, training policies for continuous control, and training large language models (LLMs) from human feedback.

## 2 RELATED WORK

### 2.1 RL AS INFERENCE

Viewing reinforcement learning through the lens of probabilistic inference offers an alternative framing of RL (Dayan & Hinton, 1997). This "RL as inference" perspective has gained considerable attention recently (Levine, 2018) inspiring various expectation-maximization (EM) based RL algorithms (Peters et al., 2010; Abdolmaleki et al., 2018). Essentially, these policy improvement algorithms can be viewed as performing EM to optimize the likelihood of a successful outcome. However, a limitation of these algorithms is their reliance on successes (preferred outcome) data. In this paper, we extend this framework to incorporate dis-preference information; effectively allowing the policy to make unwanted outcomes *less* likely. We show that this alone can have an positive effect on data efficiency and performance on certain tasks, notwithstanding the added flexibility.

### 2.2 PREFERENCE OPTIMIZATION

Preference optimization methods like Direct Preference Optimization (DPO; Rafailov et al., 2023) and Identity Preference Optimization (IPO; Azar et al., 2023) have enjoyed much attention lately, especially in the LLM training literature. This success is mostly due to a so-called *direct* optimization of human preferences, in contrast to reward model training required in RL from human feedback (RLHF) training pipelines. Nevertheless, these preference optimization methods were designed specifically to learn from a particular type of data: pairs of preferred and dis-preferred data, usually coming from humans indicating their preference over a pair of LLM responses to their query. This can be restrictive in scenarios where multiple outcomes need to be considered, and DPO has since been extended to multiple generations and compared to a novel method Efficient Exact Optimization (EXO; Ji et al., 2024), both shown to outperform the RLHF baseline in cases where a reward model is available. In this paper, we leverage the RL as inference framework to generalize preference optimization even further, allowing for more general algorithms derived from first principles. Our approach can not only handle scenarios with multiple generations but it also naturally handles cases where only one type of feedback is accessible (i.e. all generations are failures), which can be particularly useful for challenging task with binary success/failure outcomes (e.g. code, math, safety assurance).

## 3 USING POSITIVE AND NEGATIVE FEEDBACK FOR POLICY OPTIMIZATION

In this section we present an approach to optimising policies based on preference data. We build upon a large body of existing work in probabilistic inference for policy optimization.We will show that, when applied to preference optimization, the Expectation-Maximization (EM) approach results in a natural formulation of maximizing (weighted) likelihood of positive outcomes. Since such a formulation is appealing due to its simplicity but cannot effectively use information about negative outcomes, we finally derive a simple extension that enables the use of dis-preferred/negative datapoints.

The resulting algorithm has multiple intriguing properties: it can make use of preference data containing positive and negative outcomes but it does not require paired outcomes (i.e. it can make

use of data for which we only know whether it is either good or bad, without knowing about relative preference with respect to other data-points) and can thus also naturally utilize unbalanced datasets (where e.g. we have multiple preferred options for each dis-preferred example, or vice-versa). Due to the close relationship of our algorithm to the existing MPO algorithm (Abdolmaleki et al., 2018) *we refer to it as preference based MPO (PMPO)*. The final update rule is presented in equation 10.

### 3.1 BACKGROUND ON MAXIMISING FOR PREFERRED OUTCOMES

We review the preference based RL formulation common in RLHF (Ziegler et al., 2019; Rafailov et al., 2023) and show how methods from the literature on EM based policy optimization (Rawlik et al., 2013; Peters et al., 2010; Abdolmaleki et al., 2018) can be naturally applied to it.

In the following, $x$ denotes the conditioning variable such as the state/observation in classical RL, or a document and query in the LLM finetuning setting. Providing this information to a model (or policy) produces $\pi(y|x)$, a probability distribution over outputs $y$; these would be actions in classical RL or responses/generations (sequences of tokens) in the LLM finetuning literature. We will also make use of the definition of a KL divergence between conditional distributions which we define as $\mathrm{KL}(p(\cdot|x) \| q(\cdot|x)) = \mathrm{KL}(p, q; x) = \mathbb{E}_{y \sim p(\cdot|x)}[\log p(y|x) - \log q(y|x)]$.

**Objective.** Define a binary random variable $S$, which takes a value of $1$ in the event of a preferred/successful outcome and $0$ otherwise. To lighten notation, we will use the shorthand $p(S)$ and $p(S')$ to mean $p(S = 1)$ and $p(S = 0)$, respectively, and similarly for the conditioned distributions. In words, our goal is to optimize the parameters $\theta$ of a parametric policy $\pi_\theta(y|x)$ to produce outcomes $y$ that have a high likelihood of being preferred as measured by a likelihood function $p(S|y, x)$, i.e. we optimize the expected likelihood that $y$ is a 'preferred' or 'successful' response to the condition $x$:

$$\max_\theta \mathbb{E}_{y \sim \pi_\theta} p(S|y, x) \qquad (1)$$

**Reference model.** In addition to the above general formulation we assume access to a reference model $\pi_{\mathrm{ref}}$ that can either consist of a previous iteration of the model we would like to improve, or be the outcome of a pre-training or supervised fine-tuning phase (as routinely employed in RLHF for LLMs). We refer to this model as the reference policy and in general we use the terms model and policy interchangeably.

**Preference information.** In order to derive a practical sample based algorithm we assume knowledge of the likelihood function $p(S|y, x)$ to evaluate the samples $y \sim \pi_{\mathrm{ref}}(\cdot|x)$. We distinguish two cases in this paper. In the first case, this can be achieved through an evaluation function $f(y, x)$ that assigns higher values to preferred responses $y$. We can then define likelihoods as $p(S|y, x) \propto \exp(f(y, x)/\eta)$ (for preferred) and $p(S'|y, x) \propto \exp(-f(y, x)/\eta')$ (for dis-preferred) $\eta$ with temperature parameters $\eta$ and $\eta'$. The evaluation function $f$ can be derived from available reward functions $r$ or state-action value functions $Q$[1]. Alternatively, preference information can be obtained from a dataset of labeled examples:

$$\mathcal{D} = \left\{ x^{(i)}, \ y^{(i,j)}, \ s^{(i,j)} \right\}_{i,j=1}^{N,M}$$

where $s^{(i,j)}$ are binary (zero or one) preference labels (preferred or dis-preferred) usually obtained from human feedback for samples $y^{(i,j)} \sim \pi_{\mathrm{ref}}(\cdot|x^{(i)})$. In this case we are assuming $p(S|y^j, x^i) = s^{(i,j)}$ and $p(S'|y^j, x^i) = 1 - s^{(i,j)}$. Ultimately, our algorithm assumes access to information regarding the likelihood of preferred or dis-preferred events for samples drawn from $\pi_{ref}$. Note that defining the likelihood function is a design choice and depends on the information available.

**Policy optimization.** Let us drop the superscripts $(i)$ for now and only consider the objective on a per-condition basis, ultimately we average over the batch. Then for every conditioning $x = x^{(i)}$, the problem is finding a policy $\pi(y|x)$ that achieves the highest expected probability of preferred outcomes for a condition $x$. This amounts to optimizing

$$\max_\pi \log[\mathbb{E}_{y \sim \pi} p(S|y, x)] = \underbrace{\mathbb{E}_{y \sim q} \left[ \log \frac{\pi(y|x) p(S|y, x)}{q(y|x)} \right]}_{\mathcal{J}(\pi; q, x)} + \mathrm{KL}(q(y|x) \| p_\pi(y|S, x)), \qquad (2)$$

---

[1] Defined as $Q(y, x) = \mathbb{E}[\sum_t \gamma^t r(y_t, x_t)|x_o = x, y_0 = y]$ for a timeseries of observation/action pairs.

where we have used a standard formulation from the probabilistic inference literature (Kingma & Welling, 2013) to decompose the objective into an evidence lower bound $\mathcal{J}(\pi; q, x)$ and a KL term by introducing an auxiliary variational distribution $q$ [2]. The goal of EM is to iteratively find a tight lower bound given the current estimate $\pi_{\text{ref}}$ by optimizing for $q$ (E-Step) and improve the lower bound $\mathcal{J}$ by optimizing for $\pi$ (M-Step). More concretely, in the E-step, we fix $\pi = \pi_{\text{ref}}$ and find the $\hat{q}$ which minimizes the KL; this tightens the bound. In the M-step, we fix $q = \hat{q}$ and maximize the lower bound $\mathcal{J}(\pi_\theta; \hat{q}, x)$ to update $\pi_\theta$. This process of tightening the bound and improving the policy constitutes one iteration of policy improvement over $\pi_{\text{ref}}$.

**E-step:** Tighten the lower bound by fixing $\pi = \pi_{\text{ref}}$ and minimize $\text{KL}(q(\cdot|x) \parallel p_{\pi_{\text{ref}}}(\cdot|S, x))$. Following prior work (Dayan & Hinton, 1997; Peters et al., 2010; Abdolmaleki et al., 2018), since the KL is minimized when both distributions are equal, the solution can be expressed in closed form as $\hat{q}(y|x) = p_{\pi_{\text{ref}}}(y|S, x)$. Then, according to Bayes rule:

$$p_{\pi_{\text{ref}}}(y|S, x) = \frac{1}{Z_x} \pi_{\text{ref}}(y|x) p(S|y, x), \tag{3}$$

where we used the normalization factor $Z_x = \int \pi_{\text{ref}}(y|x) p(S|y, x) \, \mathrm{d}y$. Recall that likelihood function $p(S|y, x)$ is still a modelling choice discussed in the *Preference information* section.

**M-Step:** Optimize the lower bound $\mathcal{J}$ fixing $q = \hat{q}$ from the previous step. Since this problem does not have an analytic solution we use a parametric function approximator $\pi_\theta$, usually a large neural network, and maximize the following objective via gradient ascent:

$$\mathcal{J}(\pi_\theta; \hat{q}, x) = \mathop{\mathbb{E}}_{y \sim \hat{q}} \left[ \log \frac{\pi_\theta(y|x) p(S|y, x)}{\frac{1}{Z_x} \pi_{\text{ref}}(y|x) p(S|y, x)} \right] = \mathop{\mathbb{E}}_{y \sim \hat{q}} \left[ \log \pi_\theta(y|x) \right] + K \tag{4}$$

$$\mathcal{J}(\pi_\theta; x) = \mathop{\mathbb{E}}_{y \sim \pi_{\text{ref}}} \left[ \frac{p(S|y, x)}{Z_x} \log \pi_\theta(y|x) \right], \tag{5}$$

where $K$ represents all constant terms that are independent of $\theta$ and are dropped from the final objective. Notice that this objective amounts to a weighted maximum likelihood with preferences determining the weights and samples coming from $\pi_{\text{ref}}$. Notice also that the final expression subsumes the closed form E-step solution such that we can safely consider only this objective and introduce the short-hand $\mathcal{J}(\pi_\theta; x)$, dropping the implicit dependence on the E-step solution. In practice, to optimize this objective we need to form a Monte-Carlo approximation of the expectation in Eq. (5). We distinguish the two cases mentioned in the *Preference information* section.

In the first case, we assume access to a function $f$ that is proportional to the preference log-probability, and access to $M$ responses $y^{(j)}$ for each $x$. We can then set $p(S|y, x) \approx w^{(j)} \propto \exp(f(y^{(j)}, x)/\eta)$ in Eq. (5) (a softmax of $f$ across the responses $y^{(j)}$ to $x$). This is the case commonly studied in the literature, e.g., in MPO where one uses $f = Q(y^{(j)}, x)$.

It is often unrealistic to assume access to a reliable model of preference labels. For example, preferences often come from human annotations and we thus only have access to samples or we might only have access to a learned and unreliable preference model.[3] To cover this case, let us partition our dataset of labeled examples $\mathcal{D} = \mathcal{D}_a \cup \mathcal{D}_r$ where $\mathcal{D}_a = \{y^{(j)} \ni (s^{(j)} = 1)\}_{j=1:M}$ and $\mathcal{D}_r = \{y^{(j)} \ni (s^{(j)} = 0)\}_{j=1:M}$, denote accepted (preferred) samples and rejected (dis-preferred) samples, respectively. In this case we can still use the objective from Eq. (5), using the binary preferences $s^{(j)}$ as weights:

$$\mathcal{J}(\pi_\theta; x) \approx \mathcal{J}_a(\pi_\theta; x) = \mathop{\mathbb{E}}_{y^{(j)} \sim \mathcal{D}} \left[ s^{(j)} \log \pi_\theta(y^{(j)}|x) \right] = \mathop{\mathbb{E}}_{y^{(j)} \sim \mathcal{D}_a} \log \pi_\theta(y^{(i)}|x), \tag{6}$$

which effectively filters rejected generations $\mathcal{D}_r$ out, thus reverting back to the maximum likelihood objective on preferred data.

---

[2]We use the identity $\log p(X) = \int q(Z) \log \frac{p(X, Z)}{q(Z)} + \text{KL}(q(Z)|p(Z|X))$ to obtain the decomposition .

[3]A case studied in the offline RL literature where the authors realised that using binary weights often works better as in binary CRR (Wang et al., 2020).

## 3.2 USING DIS-PREFERRED OUTCOMES VIA REGULARISED MINIMUM LIKELIHOOD

We will now derive a simple way to incorporate negative (dis-preferred) samples into the optimization to address the shortcomings of naively applying the EM-based perspective from the previous section. We would like to incorporate these examples without changing the overall objective since it has well established policy improvement guarantees (Rawlik et al., 2013; Abdolmaleki et al., 2018). To accomplish this we take a second look at the non-parametric variational distribution $\hat{q}$ from Eq. (3) that is the solution to our E-step; since it determines the sampling distribution used for the M-step.

We can realise that the restriction to positive/preferred samples stems from the fact that we express $\hat{q}$ directly in terms of likelihood of preferred event $p(S|y, x)$. A natural question then is: can we re-express $\hat{q}$ in terms of dis-preferences? It turns out the answer to this is positive. Recall that $S'$ denotes the complement of the event $S$ i.e. the event that $y$ is not a successful action/response to a conditioning $x$. Then by definition, $p(S|y, x) = 1 - p(S'|y, x)$ we can equivalently write

$$\hat{q}(y|x) = \frac{1}{Z_x} \pi_{\text{ref}}(y|x)(1 - p(S'|y, x)). \tag{7}$$

We can plug this form of $\hat{q}$ into the evidence lower bound expressed in Eq. (4). After rearranging terms and re-writing in terms of two expectations over $\pi_{\text{ref}}$ this gives the alternative form:

$$\mathcal{J}(\pi_\theta; x) = \underset{y \sim \pi_{\text{ref}}}{\mathbb{E}} \left[ -\frac{p(S'|y, x)}{Z'_x} \log \pi_\theta(y|x) \right] - \frac{1}{Z'_x} \text{KL}(\pi_{\text{ref}}, \pi_\theta; x) + K, \tag{8}$$

where $K$ again denotes terms independent of $\pi_\theta$ and we used the normalization factor $Z'_x = \int \pi_{\text{ref}}(y|x)p(S'|y, x)\,\mathrm{d}y$. This version of the objective now is expressed in terms of the likelihood dis-preference event. Additionally the state-dependent constant $\frac{1}{Z'_x}$ weights the KL term on a per-state basis. This weighting implies that if the reference policy has more negative than positive examples for state $x$, the KL weight should be lower, permitting greater deviation from the reference policy. Conversely, if there are fewer negative examples, the KL weight should be higher, preserving positive examples. We simplify this by using a state-independent parameter $\beta$ to subsume this weighting. We now write the equation 8 based on dis-preferred examples:

$$\mathcal{J}(\pi_\theta; x) \approx \mathcal{J}_r(\pi_\theta; x) = \underset{y^{(j)} \sim \mathcal{D}_r}{\mathbb{E}} \left[ -\log \pi_\theta(y^{(j)}|x) \right] - \beta \, \text{KL}(\pi_{\text{ref}}, \pi_\theta; x). \tag{9}$$

where $\beta$ should be tuned and set high enough to only remove the dis-preferred samples from the prior $\pi_{\text{ref}}$ and retain the preferred samples. As before, our use of samples $s^{(j)}$ (labelled data) filters out part of the dataset; in this case, it is the accepted responses which are filtered out, hence the expectation over $\mathcal{D}_r$. We refer to the Appendix C for a full derivation. This is a fairly intuitive objective to optimize. It tells us to *minimize* the likelihood of dis-preferred examples while staying close to the reference model. Interestingly, compared to the preferred data case, it has an additional KL term that appears as a result of the reparameterization of the variational distribution. We will see in the experiments that this term is required when learning from negative data. Intuitively, we can think of the objective as modifying the reference distribution such that the negative examples are removed. Interestingly such an additional KL for the M-step has previously been considered in the literature even for the case where we only learn from positive feedback (Abdolmaleki et al., 2018). However, previous work used the additional KL term to prevent rapid entropy loss. In contrast, our motivation for incorporating the KL term is to learn from negative samples, as suggested by the derivations.

## 3.3 LEARNING FROM PREFERRED AND DIS-PREFERRED OUTCOMES

Finally, we can form a combined objective from our two M-step estimates – which both optimize the same quantity but can utilize different samples. That is, we combine Eq. (6) and Eq. (9):

$$\mathcal{J}_{ar}(\pi_\theta; x) = \underbrace{\alpha \underset{y \sim \mathcal{D}_a}{\mathbb{E}} \left[ \log \pi_\theta(y|x) \right]}_{\text{Learning From Accepted/Positive Samples}} \underbrace{-(1 - \alpha) \underset{y \sim \mathcal{D}_r}{\mathbb{E}} \left[ \log \pi_\theta(y|x) \right] - \beta \, \text{KL}(\pi_{\text{ref}}, \pi_\theta; x)}_{\text{Learning From Rejected/Negative Samples}}, \tag{10}$$

---

[4]Note that, based on the derivations, $\beta$ should approach zero as $\alpha$ approaches one. However, for cleaner comparisons in the experiments, we keep the $\alpha$ and $\beta$ parameters independent. We will empirically show that, as suggested by the derivations, no KL term is needed ($\beta = 0$) when learning only from positive examples ($\alpha = 1$) while the KL term is indeed necessary for learning from negative examples.

Figure 1: Performance of PMPO and DPO on Benchmark Functions - This figure illustrates the optimization progress of PMPO variants (PMPO-AR, PMPO-A, PMPO-R) on a selection of standard benchmark functions, showcasing their ability to leverage different types of preference feedback.

where $\alpha$ is a trade-off parameter between the two estimates. Recall that in practice, this objective will be aggregated over an entire dataset of conditions $x$ and corresponding datasets $\mathcal{D}_a$ and $\mathcal{D}_r$. There are a few interesting things to note about this objective. First, we emphasize that our objective assumes categorization of samples into good/bad or preferred/dis-preferred datasets. As a result, it can be used *even when only positive or only negative samples are available* (this is in contrast to e.g. DPO (Rafailov et al., 2023) or IPO (Azar et al., 2023) which require relative scores of paired positive and negative examples for each query $x$). Furthermore, the objective has also no restriction on the number of positive / negative samples per query $x$ and thus it automatically extends to the multi-sample case for fine-tuning language models. Finally, the objective is intuitive and simple to implement; it amounts simply to maximizing likelihood of good data while minimizing likelihood of bad data and staying close to the reference model. PMPO introduces $\alpha$ and $\beta$ hyperparameters(see Appendix A for tuning guidance). Furthermore the KL term is implemented in closed form whenever possible. For example, for the autoregressive models used in LLMs, we use the sum of per-token closed-form KL divergences of categorical distributions. This enable us to learn only from a negative feedback without access to positive feedback as suggested by our derivations. See Appendix B for KL computation details and a discussion on the importance of closed-form KL computation..

## 4  EXTRACTING PREFERENCES FROM EVALUATION FUNCTIONS

Our algorithm requires access to preference information, which can come directly from human feedback or be extracted from an evaluation function. This section describes the latter. We consider improving policies within a traditional reinforcement learning (RL) setting; bandit optimization and optimization of language models via RLHF. In each setting our preference-based update rule can be used in the policy improvement step. For this we need to extract preference information from a (possibly learned) evaluation function. This can be achieved in the following way:

**Generate Samples:** For a given input or state $x$, sample one or multiple generations $y$ from the current reference policy $\pi_{\text{ref}}$.

**Evaluate Actions:** Calculate the evaluation function $f(x, y)$ (e.g. a reward model in RLHF) for each input-generations pair $(x, y)$.

**Classify Actions:** If $f(x, y) \geq b(x)$, classify the generation $y$ as preferred in state $x$. Otherwise $(f(x, y) < b(x))$, classify it as dis-preferred. $b(x)$ is a state dependent baseline that is typically used to calculate advantage values. Typically in RL average reward is used as baseline.

## 5  EXPERIMENTS

We evaluate our algorithm in a variety of different settings, showcasing its utility as a general preference optimization algorithm that can deal with many different forms of preference feedback. In this section, we aim at confirming our derivations in practice to learn from only negative feedback, only positive feedback, or both. We first test it in a Bandit setting (optimizing synthetic benchmark functions) then in a setting where we transform RL on control and robotics tasks into preference optimization. And finally we showcase strong performance for RLHF of large language models. To verify our derivations, we evaluate three different variants of the PMPO algorithm: learning only from accepted samples ($\alpha = 1$), learning only from rejected samples ($\alpha = 0$), and learning from both accepted and rejected samples ($\alpha = 0.5$). We also use MPO (Abdolmaleki et al., 2018) and DPO (Rafailov et al., 2023) as baselines. For all the experiments, we will use a beta value of 0.5 for learning

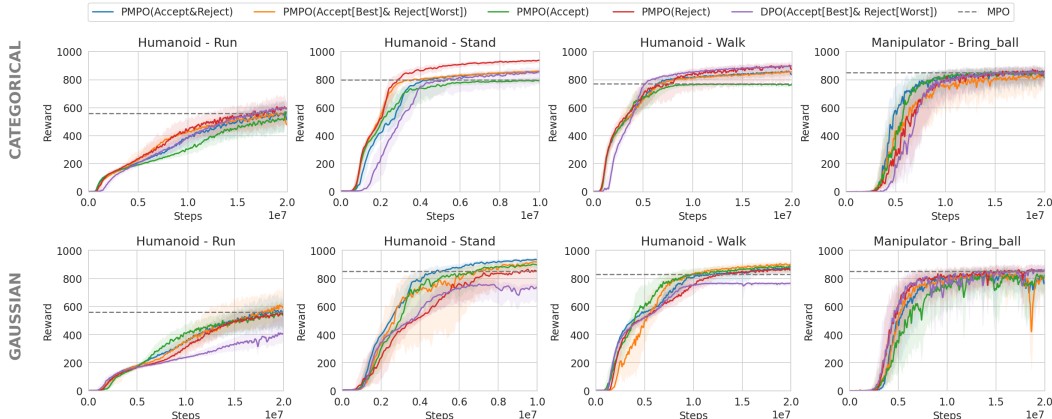

Figure 2: Comparison of PMPO/DPO/MPO for high-dimensional control tasks from the DeepMind Control Suite. We plot average reward over time of training (using 100 episodes for each evaluation).

from accept&reject, 0.0 for learning from accept only, and 2.0 for learning from reject only, unless stated otherwise. Furthermore, in all experiments except experiment 5.3, the reference policy for all baselines is updated every N steps to allow for multiple policy improvement steps and demonstrate that our algorithm can effectively optimize the underlying reward function until convergence. For experiment 5.3, we only have access to samples from the reference policy; therefore, we can make only one improvement step, which means the reference policy is effectively fixed. Please note that experiment 5.3 is designed to have access to only positive or negative feedback for each state.

## 5.1 BANDIT RL: STANDARD FUNCTIONS

Our algorithm is first evaluated on synthetic benchmarks: Rosenbrock, Sphere, and Schwefel functions (Hansen et al., 2003) with optimum value of zero, framed as multi-armed bandits (Auer et al., 2002) (no state conditioning, $x$). Per iteration, the policy samples 4 examples, receiving function values. The top 2 samples are labeled as preferred, the others dis-preferred. Figure 1 shows PMPO's performance with different feedback (PMPO-AR: all 4 labeled samples, PMPO-A: accepted only, PMPO-R: rejected only). All variants optimize the functions, demonstrating effective use of diverse preference information. Remarkably, PMPO-R (negative samples only) optimizes with the KL constraint. DPO (best/worst samples) performs similarly to PMPO-AR.

## 5.2 FULL ONLINE RL: CONTROL SUITE

We evaluate our algorithm on a range of control tasks from the DeepMind Control Suite (Tunyasuvunakool et al., 2020). See Appendix E for details. We cast the setting of optimizing a policy for the control suite as a preference optimization problem by leveraging a learned action-value function (a Q-function)–represented by a separate network trained alongside the policy–to infer preferences for each observed state and action. This is analogous to the actor-critic setting in classical reinforcement learning. Similar to the bandit case, at each iteration, the reference policy proposes four actions for each state in the batch. The top two actions with the highest Q-values are considered preferred samples, while the two actions with the lowest Q-values are treated as dis-preferred samples. We consider two different cases, one where the output of the neural network are mean and standard deviation of a Gaussian control policy and one where the actions are discretized into bins (and the network outputs categorical logits over these bins).

Figure 2 demonstrates that, as in the bandit case, our algorithm can effectively learn from different types of available signals (accept/reject, accept-only, reject-only) to solve high-dimensional tasks, such as controlling humanoid agents to run, stand, and walk, as well as manipulating objects. In all of them PMPO matches or outperforms the strong MPO baseline. Notably, even with only reject signals (PMPO-R), the algorithm is capable of achieving good performance. As predicted by the theory, not using a KL can quickly lead to collapse when using only dis-preferred samples. We also compare to an implementation of DPO (Rafailov et al., 2023) which uses the best and worst action sample

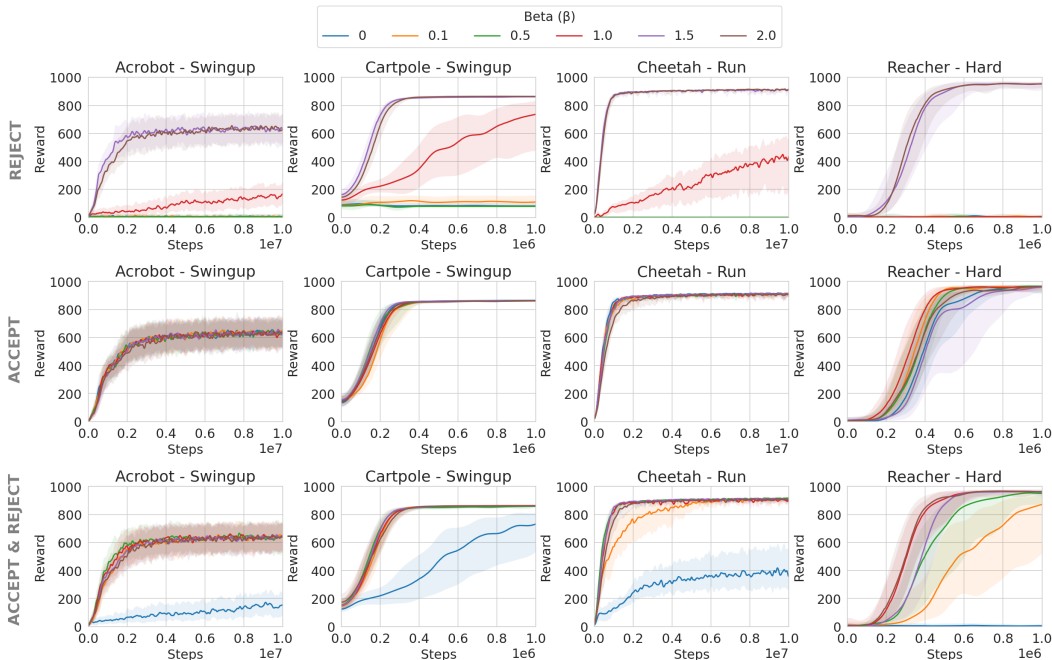

Figure 3: Impact of the KL weight 'beta' on the performance of PMPO. When learning solely from dispreferences across various Control Suite tasks (Reject, $\alpha = 0$), a sufficiently high beta value is required for effective learning. However, when learning from preferences only (Accept) PMPO is robustness to the KL weight 'beta' across different Control Suite tasks, confirming theoretical insights. When both both accept and reject signals are used (Accept & Reject), PMPO shows a partial sensitivity to KL Weight 'beta'. While learning is possible with a wider range of beta values, a beta higher than 0.5 is generally needed for optimal performance.

among the 4 samples. This still results in a strong algorithm that works well when using a discretized action representation. However, in the continuous Gaussian case, DPO requires a very high implicit regularization parameter ($\beta = 20$) which results in slow learning and suboptimal policies. For the sake of fair comparison with DPO that uses the worst and best generation, we also show results for PMPO when only the best is labeled as preferred and the worst is labeled as dispreferred, which is still competitive with DPO. Also see Appendix F for further comparisons.

We further ablate the impact of the KL term on learning solely from dispreferences ($\alpha = 0$), solely from preferences ($\alpha = 1$), and from both ($\alpha = 0.5$). For each of these settings, we sweep over the $\beta$ parameter in the range $(0.0, 0.5, 1.0, 1.5, 2.0)$. As depicted in Figure 3, when learning exclusively from dispreferences (PMPO-R), the performance is highly sensitive to $\beta$. To achieve effective learning, we need to set $\beta$ sufficiently high ($> 1.0$), which aligns with our theoretical derivations. In contrast, Figure 3 shows that the algorithm is insensitive to the setting of $\beta$ when learning only from preferred samples (PMPO-A), again confirming our theoretical insights. When learning from both types of signals (PMPO-AR), as shown in Figure 3, we observe a partial sensitivity to the KL weight $\beta$. While the algorithm can learn with a wider range of beta values, a $\beta$ larger than 0.5 is still necessary to ensure optimal performance across all tasks.

## 5.3 OFFLINE RL USING ADVANTAGE FUNCTION

In a final set of experiment on control domains we want to show that our algorithm can also be applied to a setting where we have only access to one sample with either a reject or an accept label per state conditioning $x$. We consider the RGB Stacking benchmark (Lee et al., 2021), a pick-and-place manipulation task with image observations (see Appendix E for details). We investigate the effect of positive and negative feedback in the context of offline RL to exclude cascading effects from exploration. To this end we take a dataset of 140k episodes from a multi-task RL experiment trained to convergence (Lampe et al., 2024). We then train a value function on all data and use it

to label the transitions in the first 40k episodes as accept (positive advantage) or reject (negative advantage). Different combinations of acceptance, rejection, and BC losses are then compared in order to understand their respective effects. In summary we use: i) the full 140k episodes to train a value function and label the first 40k episodes as accept or reject ; ii) the first 40k episodes to compute the positive weighted part of the loss if labeled as accept and to compute the negatively weighted part of the loss if labeled as reject. The KL part of the loss is calculated on all 140k episodes. Note that the value function is only used to transform the reward annotations into accept and reject labels.

Table 1 shows the achieved reward for different loss combinations. First we run BC on the full 140k episodes and we can observe that the performance is mediocre due to the data containing a significant amount of bad episodes. Using only the accepted transitions for BC training does not result in better performance; this is due to the limited number of positive examples contained in the first 40k episodes. When combining both BC and using the positive (accept) part of the loss, performance does not significantly improve as the large number of negative episodes is not compensated for. On the other hand, combining BC with the negative (reject) part of the loss does significantly improve performance. This is due to the rejection loss successfully pushing the policy away from the negative examples (while keeping close on all other data due to the KL constraint). Finally, best performance is achieved when combining all three losses; and thus effectively utilizing all data. While in this example we have constructed the dataset in a way that the effect is strong, and this might be less the case in more natural settings, it nevertheless shows that using a negative signal can have a significant effect on performance by masking the effect of bad data.

|  | BC | Accept+BC | Accept | Reject+BC | Accept+Reject+BC |
|---|---|---|---|---|---|
| Reward | 24 | 26 | 27 | 77 | 93 |

Table 1: Comparing different mixtures of acceptance, rejection and BC losses. We measure average reward (over 100 evaluation episodes) across stacking of all 5 triplets. Training with BC is corrupted by bad examples. Training on only accepted examples lacks data. Only when integrating the rejection loss bad data can be masked and performance goes up. Best performance is achieved when combining acceptance, rejection and BC loss signals.

## 5.4 LANGUAGE ALIGNMENT EXPERIMENTS

We apply different versions of the PMPO algorithm to the task of aligning large language models. Specifically, we fine-tune a Gemma 2B pre-trained model using a trained reward model (Team, 2024b) using prompts from the LMSYS-chat-1M dataset (Zheng et al., 2023). The reward model has been trained on human preference data with a Bradley-Terry modelisation as explained in (Christiano et al., 2017). In these experiments, we perform one epoch of training, processing a dataset of 500k prompts in approximatively 4000 learner steps, meaning that each batch is composed of 128 prompts and 4 generations per prompt. Similar to the typical RLHF setting, at each iteration, for each prompt in a batch, we sample four generations from the model and rank them based on their reward values. The top two generations are labeled as preferred, and the bottom two as dis-preferred. For the sake of fair comparison with DPO that uses the top one (best) and bottom one (worst) generation, we also show results for PMPO when only the top one is labeled as preferred and the bottom one is labeled as dispreferred. Note that this particular choice could be refined further and tailored to the task. First, Fig. 4 showcases the best PMPO setting, leveraging both accept and reject signals (PMPO-AR) (and we compare to use either feedback signal in isolation). Notably, utilizing both types of feedback leads to faster learning compared to using either signal in isolation (PMPO-A or PMPO-R) and overall our approach is competitive to DPO, which is applicable in this setting by using only the best and worst sample respectively per prompt but would be more restrictive in general (i.e. it cannot naturally make use of unbalanced preference data). As shown on the right, when performing a side by side comparison using GPT-4 (OpenAI, 2024) to judge whether our model is preferred over the base Gemma model (using a set of held-out test prompts) the PMPO fine-tuned model wins over the base model. Note in Fig. 4 right, we see some drop indicating some exploitation of the imperfect reward model; known as reward hacking (Skalse et al., 2022). We can see that PMPO-AR is the quickest to "hack the reward"(see Fig. 4 left), it reaches a good performance but then in the middle of training its start hacking the reward and learns a pathological behaviour that makes it performs worse on the independent benchmark. This phenomenon has been observed consistently in RLHF. Overall, our language alignment experiments provide strong evidence for the effectiveness and versatility of

PMPO. Finally, we illustrate in Figure 5 that, again, our algorithm demonstrates the ability to learn effectively from various preference signals, including scenarios with only accept (PMPO-A), only reject (PMPO-R), or both accept/reject (PMPO-AR) feedback. These results highlight the versatility of our approach to different preference acquisition settings. The results also underline the critical role of the KL term in enabling learning exclusively from dis-preferred generations (PMPO-R). As predicted by our derivation, a sufficiently high value $\beta > (1 - \alpha)$ is necessary to stabilize learning in this scenario. In contrast, when learning solely from preferred samples (PMPO-A), the algorithm is insensitive to the value of $\beta$ in terms of stability.

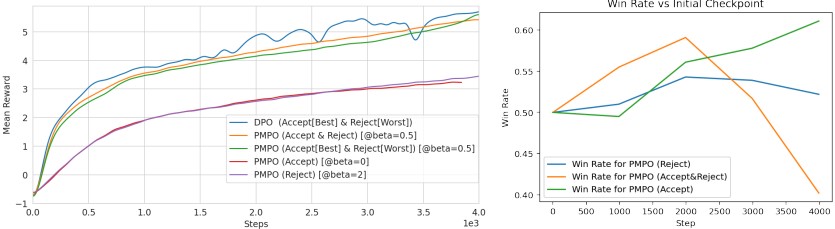

Figure 4: Left: Impact of Combining Accept and Reject Signals - The plot demonstrates the learning progress of PMPO-AR (using both accept and reject signals) compared to PMPO-A and PMPO-R, showcasing faster learning when leveraging both types of feedback in language alignment task and is competitive with DPO. Right: Win-rate when doing A/B comparisons on held-out prompts for PMPO against the base Gemma checkpoint as judged by GPT-4.

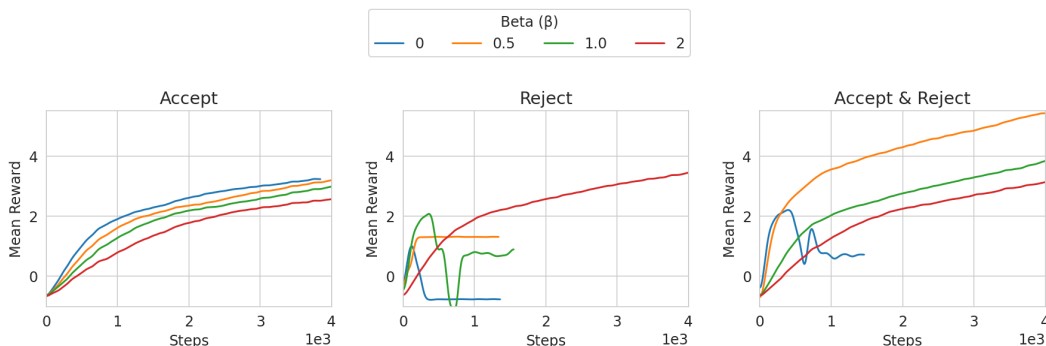

Figure 5: Rewards obtained by the policy at each training step, averaged over the batch and smoothed. Each curve corresponds to a configuration of $\beta$ specified in the legend. This figure illustrates the ability of PMPO to learn effectively from various preference signals (accept-only, reject-only, or both) in language alignment tasks. highlighting its adaptability to different preference acquisition settings.

## 6 CONCLUSION

We propose a novel algorithm for policy optimization from preference feedback derived from the perspective of RL as probabilistic inference. Our policy improvement algorithm has a clear and intuitive objective: it maximizes the likelihood of preferred data while minimizing the likelihood of dis-preferred data. We show that doing the latter in a stable way requires a regularization term forcing the policy to stay close to a reference model. This regularization term follows naturally from the derivation. The main advantage of our algorithm over existing preference optimization algorithms such as DPO is that it does not rely on defining/fitting an explicit model of the preferences and can thus use data containing partial preference information; i.e. we can use data where instead of comparisons between samples we only have accept (or only reject) labels and make no further assumptions on their distribution. One limitation of our approach is that to effectively learn from negative feedback, we need a good estimate of the KL term, ideally in closed form; otherwise, enough samples from the reference model are needed.

## 7 ACKNOWLEDGMENTS

We are grateful to the Gemma team for providing the models and infrastructure that enabled our language alignment experiments. We also thank Thomas Hubert and Markus Wulfmeier for their valuable feedback, and John Agapiou for identifying a subtle inconsistency in our derivations and providing suggestions for its resolution.

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

## A    Hyper parameter tuning

PMPO introduces two hyperparameters, $\alpha$ and $\beta$. However, we found them relatively easy to tune in practice based on the following guidelines:

- **$\alpha$**: This parameter reflects the relative importance of positive and negative feedback. When both are available, $\alpha = 0.5$ is a reasonable starting point if both types of feedback are equally reliable. Otherwise, can be adjusted to reflect the confidence in each feedback type. In cases with only one type of feedback, is naturally determined by the data.

- **$\beta$**: This parameter controls the influence of the prior or reference policy. Our experiments suggest that a higher is generally beneficial when learning from negative feedback such that the more is contribution of negative feedback to the policy update (lower $\alpha$) the more $\beta$ should be. This can also be motivate from the derivations that KL term is appear as the result of learning from negative feedback. We also find that as long as $\beta$ is large enough, the algorithm is fairly insensitive to the exact choice of the parameter. Please also see section 3.2 in the main paper for more insights regarding derivation of parameter $\beta$.

## B    Computing the Kullback-Leibler divergence

learning from negative feedback is only feasible with a correct KL term and access to a reference policy $\pi_{ref}$. When logits are available using an exact KL computation is ideal. In the absence of logits (Experiment 5.3), we relied on an abundance of unlabelled data to estimate it. This enables us to learn only from negative feedback without access to positive feedback as suggested by our derivations.

Subsequently, we present the derivations for computing KL divergence for LLMs with autoregressive policies in our experiments.

Let's consider a single two-token generation $(y_1, y_2)$ sampled autoregressively from a prompt $x$. We can factorize the joint distribution associated with such a generation as follows:

$$\pi(y_1, y_2|x) = \pi^1(y_1|x)\pi^2(y_2|x, y_1) \tag{11}$$

Using this factorization for both $\pi_\theta$ and $\pi_{\text{ref}}$ in our KL regularizer, we compute the following:

$$\text{KL}(\pi_{\text{ref}}\|\pi_\theta) = \mathbb{E}_{\pi_{\text{ref}}} \log \frac{\pi_{\text{ref}}}{\pi_\theta} \tag{12}$$

$$= \mathbb{E}_{\pi_{\text{ref}}^1} \mathbb{E}_{\pi_{\text{ref}}^2} \log \frac{\pi_{\text{ref}}^1}{\pi_\theta^1} \frac{\pi_{\text{ref}}^2}{\pi_\theta^2} \tag{13}$$

$$= \mathbb{E}_{\pi_{\text{ref}}^1} \mathbb{E}_{\pi_{\text{ref}}^2} \log \frac{\pi_{\text{ref}}^1}{\pi_\theta^1} + \mathbb{E}_{\pi_{\text{ref}}^1} \mathbb{E}_{\pi_{\text{ref}}^2} \log \frac{\pi_{\text{ref}}^2}{\pi_\theta^2} \tag{14}$$

$$= \mathbb{E}_{\pi_{\text{ref}}^1} \log \frac{\pi_{\text{ref}}^1}{\pi_\theta^1} + \mathbb{E}_{\pi_{\text{ref}}^1} \mathbb{E}_{\pi_{\text{ref}}^2} \log \frac{\pi_{\text{ref}}^2}{\pi_\theta^2}, \tag{15}$$

where the first term can drop the expectation with respect to $\mathbb{E}_{\pi_{\text{ref}}^2}$ since the integrand $\log \frac{\pi_{\text{ref}}^1}{\pi_\theta^1}$ does not depend on $y_2$. The first term can be computed analytically because $\pi_{\text{ref}}^1$ and $\pi_\theta^1$ are simply categorical distributions over the entire vocabulary of tokens. The second term, however, is problematic due to the outer expectation $\mathbb{E}_{\pi_{\text{ref}}^1}$, which requires us to integrate a KL over all possible values of $y_1$. While we can easily compute this KL for any one value of $y^1$, the integration would be unwieldy and certainly becomes intractable as we extend this to longer sequences $y_3$, $y_4$, etc. Luckily, during training we obtain samples $\tilde{y}_1, \tilde{y}_2 \sim \pi_{\text{ref}}$ and so $\tilde{y}_1 \sim \pi_{\text{ref}}^1$, which allows us to use a single-sample Monte Carlo unbiased estimate of the second term such that:

$$\approx \mathbb{E}_{\pi_{\text{ref}}^1} \log \frac{\pi_{\text{ref}}^1}{\pi_\theta^1} + \mathbb{E}_{\pi_{\text{ref}}^2} \log \frac{\pi_{\text{ref}}^2(\cdot|x, \tilde{y}_1)}{\pi_\theta^2(\cdot|x, \tilde{y}_1)}. \tag{16}$$

Extending this to sequences of length $L$, and dropping the superscripts on the policies as they are autoregressive, we obtain the following approximation:

$$\text{KL}(\pi_{\text{ref}} \| \pi_\theta) \approx \mathbb{E}_{\pi_{\text{ref}}} \log \frac{\pi_{\text{ref}}(\cdot|x)}{\pi_\theta(\cdot|x)} + \sum_{i=1}^{L-1} \mathbb{E}_{\pi_{\text{ref}}} \log \frac{\pi_{\text{ref}}(\cdot|x, \tilde{y}_{1:i})}{\pi_\theta(\cdot|x, \tilde{y}_{1:i})}. \tag{17}$$

where $\tilde{y}_{1:L} \sim \pi_{\text{ref}}$.

## C  FULL DERIVATIONS OF PMPO UPDATE RULES

Consider a decision-making scenario characterized by the following elements:

**Stationary State Distribution:** A stationary distribution $\mu(x)$ describes the probability of encountering different states or contexts $x$. We typically have access to samples from this distribution.

**Policy:** A policy $\pi(y|x)$ dictates the probability of selecting an action/outcome $y$ given a state $x$. We also refer to the current estimate of the policy as the reference policy, denoted by $\pi_{ref}(y|x)$.

**Preference Information:** We have access to knowledge for driving the likelihood of preference or dis-preference toward actions taken by a policy $\pi(y|x)$. This information takes one of these forms:

- **Preference Probabilities:** $q(I = 1|y, x)$, where $I$ is a binary indicator with $I = 1$ representing preference. When only a reward or Q-function is available, this probability can be derived as $q(I = 1|y, x) = \frac{\exp(Q(y,x))}{Z_q}$.
- **Dis-preference Probabilities:** $q(I = 0|y, x)$, where $I = 0$ denotes dis-preference. When only a reward or Q-function is available, this probability can be derived as $q(I = 0|y, x) = \frac{\exp(-Q(y,x))}{Z_q}$.

where $Z_q$ is a normalization constant $Z_q = \exp(Q(y, x)) + \exp(-Q(y, x))$ to ensure $q(I = 0|y, x) + q(I = 1|y, x) = 1$. Note that $q(I = 1|y, x)$ and $q(I = 0|y, x)$ are likelihood functions and can be defined depending on the problem at hand.

**Objective:** Derive an optimal policy to give the highest probability to the preferred outcomes:

$$\max_\theta \log p_\theta(I = 1) = \log \int \mu(x) \int \pi_\theta(y|x) q(I = 1|y, x) \, dy \, dx$$

### C.1  MAXIMIZING FOR PREFERRED OUTCOMES

We start with the stated objective:

$$\max_\theta \log p_\theta(I = 1) = \log \int \mu(x) \int \pi_\theta(y|x) q(I = 1|y, x) \, dy \, dx$$

This objective can be solved by the EM algorithm to iteratively create a tight lower bound on the current estimate of the objective $\log p_{\text{ref}}(I = 1)$, and then we optimize the objective. By repeating these two steps, it is guaranteed that the algorithm will converge. We will show one iteration of EM to improve the current estimate $\pi_{\text{ref}}$. In order to do so, we use the following identity:

$$\log p(X) = \int q(Z) \log \frac{p(X, Z)}{q(Z)} dZ + \text{KL}(q(Z)|p(Z|X))$$

which gives us the following equivalent right-hand side:

$$\log p_\theta(I = 1) = \int \mu(x) \int q(y|x) \log \frac{p_\theta(I = 1, y|x)}{q(y|x)} dy dx + \int \mu(x) \, \text{KL}(q(y|x)|p_\theta(y|I = 1, x)) dx$$

### C.1.1 E-STEP

In the E-step, our goal is to choose the variational distribution $q(y|x)$ such that the lower bound on

$$\log p_{\texttt{ref}}(I = 1)$$

is as tight as possible, which is the case when the KL term in the equation above is zero at the current estimate of the policy $\pi_{\texttt{ref}}$. This simply leads to $q(y|x) = p_{\texttt{ref}}(y|I = 1, x)$, or according to Bayes' rule, we get:

$$q(y|x) = \frac{\pi_{\texttt{ref}}(y|x)p(I = 1|x, y)}{p_{\texttt{ref}}(I = 1|x)}$$

where $p_{\texttt{ref}}(I = 1|x) = \int \pi_{\texttt{ref}}(y|x)p(I = 1|x, y)dy$ is a normalizer for a given state $x$.

### C.1.2 M-STEP

In the E-step, we found non-parametric variational distributions $q(y|x)$ for $x \sim \mu(x)$ that give higher probability to preferred actions sampled from $\pi_{\texttt{ref}}$. The E-step can be seen as sample-based policy improvement of $\pi_{\texttt{ref}}$ with respect to preferences. In the M-step, we optimize the lower bound to obtain a new distribution, i.e.,

$$\max_{\theta} \int \mu(x) \int q(y|x) \log \frac{p_{\theta}(I = 1, y|x)}{q(y|x)} dy dx$$

where

$$q(y|x) = \frac{\pi_{\texttt{ref}}(y|x)p(I = 1|x, y)}{\int \pi_{\texttt{ref}}(y|x)p(I = 1|y, x)}$$

according to the derivations in the E-step. After rearranging the terms and removing the terms that do not depend on $\theta$, we get

$$\max_{\theta} \int \mu(x) \int q(y|x) \log \pi_{\theta}(y|x) dy dx$$

which effectively is a weighted maximum likelihood objective to fit improved non-parametric policies.

### C.2 INCORPORATING DIS-PREFERRED OUTCOMES FOR POLICY OPTIMIZATION

In the previous section, we showed how optimizing for preferred outcomes can lead us to useful policy update rules. However, we only incorporated preferred outcomes $q(I = 1)$ to update the policy. Now we ask this question: how can we incorporate dis-preferred outcomes to directly optimize the policy? In order to do that, we first write down the policy optimization objective we derived in terms of preferences:

$$\max_{\theta} \int \mu(x) \int q(y|x) \log \pi_{\theta}(y|x) dy dx$$

where

$$q(y|x) = \frac{\pi_{\texttt{ref}}(y|x)p(I = 1|x, y)}{\int \pi_{\texttt{ref}}(y|x)p(I = 1|y, x)}.$$

We also know that $p(I = 1|x, y) = 1 - p(I = 0|x, y)$. This is correct as $p(I|x, y)$ is a probability function over a binary random variable $I$. Therefore, the variational distribution can be rewritten in terms of dis-preferences, i.e.,

$$q(y|x) = \frac{\pi_{\texttt{ref}}(y|x)(1 - p(I = 0|x, y))}{\int \pi_{\texttt{ref}}(y|x)(1 - p(I = 0|x, y))}.$$

Substituting this into the maximum likelihood term above, we get:

$$\max_\theta \int \mu(x) \int \frac{1}{1 - Z'(x)} \pi_{\texttt{ref}}(y|x) \log \pi_\theta(y|x) \, dy \, dx - \int \mu(x) \int \frac{1}{1 - Z'(x)} \pi_{\texttt{ref}}(y|x) p(I = 0|x, y) \log \pi_\theta(y|x) \, dy \, dx$$

where $Z'(x) = \int \pi_{\texttt{ref}}(y|x) p(I = 0|x, y) \, dy$.

After rearranging the terms, we get the equivalent form:

$$\max_\theta \int \mu(x) \frac{1}{1 - Z'(x)} \int \pi_{\texttt{ref}}(y|x) \log \pi_\theta(y|x) \, dy \, dx -$$
$$\int \mu(x) \frac{Z'(x)}{1 - Z'(x)} \int \frac{1}{Z'(x)} \pi_{\texttt{ref}}(y|x) p(I = 0|x, y) \log \pi_\theta(y|x) \, dy \, dx$$

which will simplify to

$$\max_\theta \int \mu(x) \int \frac{1}{Z'(x)} \pi_{\texttt{ref}}(y|x) \log \pi_\theta(y|x) \, dy \, dx - \int \mu(x) \int \frac{1}{Z'(x)} \pi_{\texttt{ref}}(y|x) p(I = 0|x, y) \log \pi_\theta(y|x) \, dy \, dx$$

first term can be written as a KL term, i.e,

$$\max_\theta - \int \mu(x) \frac{1}{Z'(x)} \text{KL}(\pi_{\texttt{ref}}(y|x)|\pi_\theta(y|x)) dx - \int \mu(x) \int \frac{1}{Z'(x)} \pi_{\texttt{ref}}(y|x) p(I = 0|x, y) \log \pi_\theta(y|x) \, dy \, dx$$

Note that $\frac{1}{Z'(x)}$ is a state-dependent constant that weights the KL term on a state-by-state basis. This constant suggests that for a state $x$, when the reference policy contains more negative examples compared to positive examples, then the KL term weight should be lower so more samples from the reference policy can be removed. Otherwise, when there are not as many negative examples to remove, then the KL term should have a high weight so positive examples remains. For simplicity, we subsume this weight into a parameter $\beta$ that is state-independent. Now the final update rule reads::

$$\max_\theta - \int \mu(x) \int t(y|x) \log \pi_\theta(y|x) \, dy \, dx - \beta \int \mu(x) \text{KL}(\pi_{\texttt{ref}}(y|x)|\pi_\theta(y|x)) \, dx$$

where $t(y|x) = \frac{\pi_{\texttt{ref}}(y|x) p(I=0|x,y)}{\int \pi_{\texttt{ref}}(y|x) p(I=0|x,y) \, dy}$ and $\beta$ is a tuning parameter. This update rule minimizes the probability of the dis-preferred distribution while staying close to the reference policy. Note that the KL term and its direction emerge directly from the derivations.

## C.3 FINAL UPDATE RULE: LEVERAGING BOTH PREFERENCES AND DIS-PREFERENCES

After putting things together, we get the following update rule that maximizes the preferred outcomes and minimizes the dis-preferred outcomes, i.e.,

$$\max_\theta \alpha \int \mu(x) \int q(y|x) \log \pi_\theta(y|x) \, dy \, dx -$$
$$(1 - \alpha) \left[ \int \mu(x) \int t(y|x) \log \pi_\theta(y|x) \, dy \, dx - \beta \int \mu(x) \text{KL}(\pi_{\texttt{ref}}(y|x)|\pi_\theta(y|x)) \, dx \right]$$

where $q$ is a distribution modified with respect to preference probabilities resulting from $\pi_{\texttt{ref}}$ weighted by preference probabilities, i.e.,

$$q(y|x) = \frac{\pi_{\texttt{ref}}(y|x)p(I = 1|x, y)}{\int \pi_{\texttt{ref}}(y|x)p(I = 1|x, y)\, dy}$$

and $t$ is a distribution modified with respect to dis-preference probabilities resulting from $\pi_{\texttt{ref}}$ weighted by dis-preference probabilities, i.e.,

$$t(y|x) = \frac{\pi_{\texttt{ref}}(y|x)p(I = 0|x, y)}{\int \pi_{\texttt{ref}}(y|x)p(I = 0|x, y)\, dy}$$

Now we can choose likelihood functions $q(I = 1|y, x)$ and $q(I = 0|y, x)$. Note that we can choose different likelihood functions depending on the problem and available information; for example, the likelihood function can depend on advantage values.

Intuitively, this objective learns from the preferred distribution $q(y|x)$ and gets away from the dis-preferred distribution $t(y|x)$ while staying close to the reference policy (which enables learning from dis-preferred distributions) .

## D FUNDAMENTAL RESULTS

This section explains precisely why the EM method is a sound approach to optimization. We will present the case of a discrete function. This section relies on classical results in discrete optimization and shows how one can build a simple strictly improving algorithm that maximises a discrete function. For simplicity and clarity, we will optimize a function $f \in \mathbb{R}^{\mathcal{S}}$ mapping elements of $\mathcal{S}$ to real numbers, where $\mathcal{S}$ is a finite set. More precisely, our goal is to find a distribution $\delta \in \Delta_{\mathcal{S}}$ (parameterized policy in RL) that maximises the expectation of $f$ under $\delta$ (expected value function in RL):

$$\mathbb{E}_{s\sim\delta}[f(s)] = \sum_{s\in\mathcal{S}} \delta(s)f(s).$$

We recall that a discrete probability distribution $\delta \in \Delta_{\mathcal{S}}$ can be identified as a positive real function $\delta \in \mathbb{R}_{+}^{\mathcal{S}}$ verifying:

$$\sum_{s\in\mathcal{S}} \delta(s) = 1.$$

To find a good distribution to maximise $\mathbb{E}_{s\sim\delta}[f(s)]$, the algorithm relies on the following results:

- Starting from a distribution $\eta \in \Delta_{\mathcal{S}}$, there exists a unique closed-form argmaximum $\delta^*$ of the regularised expectation $\mathbb{E}_{s\sim\delta}[f(s)] - \tau\text{KL}(\delta \,||\, \eta)$, where $\tau \in \mathbb{R}_{+}^{*}$ is a strictly positive real number. We have $\delta^* = \frac{\eta(\cdot)\exp(\tau^{-1}f(\cdot))}{\sum_{s'\in\mathcal{S}}\eta(s')\exp(\tau^{-1}f(s'))}$.
- Unless $\delta^* = \eta$, we have a strict improvement between $\delta^*$ and $\eta$:

$$\mathbb{E}_{s\sim\delta^*}[f(s)] > \mathbb{E}_{s\sim\eta}[f(s)].$$

We prove these results in the main paper. Those results implies that the following algorithm that starts at $\delta_0 = \eta$ and computes:

$$\forall k \in \mathbb{N}, \delta_{k+1} = \frac{\delta_k(\cdot)\exp(\tau^{-1}f(\cdot))}{\sum_{s'\in\mathcal{S}}\delta_k(s')\exp(\tau^{-1}f(s'))},$$

is strictly monotonically improving until there is $K \in \mathbb{N}$ such that $\delta_{K+1} = \delta_K$:

$$\mathbb{E}_{s\sim\delta_0}[f(s)] < \mathbb{E}_{s\sim\delta_1}[f(s)] < \cdots < \mathbb{E}_{s\sim\delta_K}[f(s)] = \mathbb{E}_{s\sim\delta_{K+1}}[f(s)].$$

In practice, we have a set of learnable weights $\theta$ to parameterize a distribution $q_\theta$ in order to fit $\delta_{k+1}$ and another set of fixed weights $\mu$ to parameterize a distribution $q_\mu = \delta_k$. Then, to fit $\delta_{k+1}$ the idea

is to minimize the following KL divergence (this is the maximisation step):

$$\mathcal{L}(\theta) = \text{KL}\left(\frac{q_\mu(\cdot)\exp(\tau^{-1}f(\cdot))}{\sum_{s'\in\mathcal{S}}q_\mu(s')\exp(\tau^{-1}f(s'))} \| q_\theta\right),$$

$$= \mathbb{E}_{s\sim\frac{q_\mu(\cdot)\exp(\tau^{-1}f(\cdot))}{\sum_{s'\in\mathcal{S}}q_\mu(s')\exp(\tau^{-1}f(s'))}}\left[\log\left(\frac{\frac{q_\mu(\cdot)\exp(\tau^{-1}f(\cdot))}{\sum_{s'\in\mathcal{S}}q_\mu(s')\exp(\tau^{-1}f(s'))}}{q_\theta}\right)\right],$$

$$= -\mathbb{E}_{s\sim\frac{q_\mu(\cdot)\exp(\tau^{-1}f(\cdot))}{\sum_{s'\in\mathcal{S}}q_\mu(s')\exp(\tau^{-1}f(s'))}}\left[\log\left(q_\theta\right)\right] + \mathbb{E}_{s\sim\frac{q_\mu(\cdot)\exp(\tau^{-1}f(\cdot))}{\sum_{s'\in\mathcal{S}}q_\mu(s')\exp(\tau^{-1}f(s'))}}\left[\log\left(\frac{q_\mu(\cdot)\exp(\tau^{-1}f(\cdot))}{\sum_{s'\in\mathcal{S}}q_\mu(s')\exp(\tau^{-1}f(s'))}\right)\right]$$

The term $\mathbb{E}_{s\sim\frac{q_\mu(\cdot)\exp(\tau^{-1}f(\cdot))}{\sum_{s'\in\mathcal{S}}q_\mu(s')\exp(\tau^{-1}f(s'))}}\left[\log\left(\frac{q_\mu(\cdot)\exp(\tau^{-1}f(\cdot))}{\sum_{s'\in\mathcal{S}}q_\mu(s')\exp(\tau^{-1}f(s'))}\right)\right]$ does not depend on $\theta$ so is irrelevant in the minimization. Using the re-weighting formula $\mathbb{E}_{s\sim\delta}[f(s)] = \mathbb{E}_{s\sim\eta}[\frac{\delta(s)}{\eta(s)}f(s)]$, the minimization problem is equivalent to:

$$\mathcal{L}(\theta) = -\mathbb{E}_{s\sim\frac{q_\mu(\cdot)\exp(\tau^{-1}f(\cdot))}{\sum_{s'\in\mathcal{S}}q_\mu(s')\exp(\tau^{-1}f(s'))}}\left[\log\left(q_\theta\right)\right],$$

$$= -\mathbb{E}_{s\sim q_\mu}\left[\frac{\exp(\tau^{-1}f(\cdot))}{\sum_{s'\in\mathcal{S}}q_\mu(s')\exp(\tau^{-1}f(s'))}\log\left(q_\theta\right)\right],$$

As $\sum_{s'\in\mathcal{S}}q_\mu(s')\exp(\tau^{-1}f(s'))$ is a constant, this is equivalent to minimizing:

$$\mathcal{L}(\theta) = -\mathbb{E}_{s\sim q_\mu}[\exp(\tau^{-1}f(s))\log\left(q_\theta(s)\right)].$$

## D.1 Existence and uniqueness of the regularized argmaximum

For completeness, we briefly recall the proof of existence and uniqueness of the argmaximum of the following regularized criterion that can also be found in the work of Rafailov et al. (2023):

$$\mathcal{L}_\tau(\delta) = \mathbb{E}_{s\sim\delta}[f(s)] - \tau\text{KL}(\delta \| \eta),$$
$$= \sum_{s\in\mathcal{S}}\delta(s)f(s) - \tau\text{KL}(\delta \| \eta).$$

Now, if we define the softmax probability $\delta^* \in \Delta_\mathcal{S}$ as:

$$\forall s \in \mathcal{S}, \delta^*(s) = \frac{\eta(s)\exp(\tau^{-1}f(s))}{\sum_{s'\in\mathcal{S}}\eta(s')\exp(\tau^{-1}f(s'))},$$

then, under the previous definitions, we have the following results:

$$\delta^* = \arg\max_{\delta\in\Delta_\mathcal{S}}\mathcal{L}_\tau(\delta)$$

*Proof.*

$$\frac{\mathcal{L}_\tau(\delta)}{\tau} = \sum_{s \in \mathcal{S}} \delta(s) \frac{f(s)}{\tau} - \mathrm{KL}(\delta \, || \, \eta),$$

$$= \sum_{s \in \mathcal{S}} \delta(s) \frac{f(s)}{\tau} - \sum_{s \in \mathcal{S}} \delta(s) \log\big(\frac{\delta(s)}{\eta(s)}\big),$$

$$= \sum_{s \in \mathcal{S}} \delta(s) \big(\frac{f(s)}{\tau} - \log\big(\frac{\delta(s)}{\eta(s)}\big)\big),$$

$$= \sum_{s \in \mathcal{S}} \delta(s) \big(\log\big(\exp(\tau^{-1} f(s))\big) - \log\big(\frac{\delta(s)}{\eta(s)}\big)\big),$$

$$= \sum_{s \in \mathcal{S}} \delta(s) \big(\log\big(\frac{\eta(s) \exp(\tau^{-1} f(s))}{\delta(s)}\big)\big),$$

$$= \sum_{s \in \mathcal{S}} \delta(s) \big(\log\big(\frac{\eta(s) \exp(\tau^{-1} f(s)) \frac{\sum_{s' \in \mathcal{S}} \eta(s') \exp(\tau^{-1} f(s'))}{\sum_{s' \in \mathcal{S}} \eta(s') \exp(\tau^{-1} f(s'))}}{\delta(s)}\big)\big),$$

$$= \sum_{s \in \mathcal{S}} \delta(s) \big(\log\big(\frac{\frac{\eta(s) \exp(\tau^{-1} f(s))}{\sum_{s' \in \mathcal{S}} \eta(s') \exp(\tau^{-1} f(s'))}}{\delta(s)}\big)\big) + \sum_{s \in \mathcal{S}} \delta(s) \log\big(\sum_{s' \in \mathcal{S}} \eta(s') \exp(\tau^{-1} f(s'))\big),$$

$$= \sum_{s \in \mathcal{S}} \delta(s) \big(\log\big(\frac{\delta^*(s)}{\delta(s)}\big)\big) + \log\big(\sum_{s' \in \mathcal{S}} \eta(s') \exp(\tau^{-1} f(s'))\big),$$

$$= -\mathrm{KL}(\delta \, || \, \delta^*) + \log\big(\sum_{s \in \mathcal{S}} \eta(s) \exp(\tau^{-1} f(s))\big).$$

By definition of the KL, we now that $\delta^* = \arg\max_{\delta \in \Delta_\mathcal{S}} \big[ -\mathrm{KL}(\delta \, || \, \delta^*) \big]$ and as:

$$-\mathrm{KL}(\delta \, || \, \delta^*) = \frac{\mathcal{L}_\tau(\delta)}{\tau} - \log\big(\sum_{s \in \mathcal{S}} \eta(s) \exp(\tau^{-1} f(s))\big) \tag{18}$$

where $\log\big(\sum_{s \in \mathcal{S}} \eta(s) \exp(\tau^{-1} f(s))\big)$ is a constant (does not depend on $\delta$) and $\tau$ a positive multiplicative term, then $-\mathrm{KL}(\delta \, || \, \delta^*)$ and $\mathcal{L}_\tau(\delta)$ share the same argmaximum. This concludes the proof. $\qquad\square$

The fact that we have:

$$\delta^* = \frac{\eta(\cdot) \exp(\tau^{-1} f(\cdot))}{\sum_{s' \in \mathcal{S}} \eta(s') \exp(\tau^{-1} f(s'))} = \arg\max_{\delta \in \Delta_\mathcal{S}} \mathcal{L}_\tau(\delta),$$

implies by simply replacing $\delta$ by $\delta^*$ in equation (18) that:

$$-\mathrm{KL}(\delta^* \, || \, \delta^*) = \frac{\mathcal{L}_\tau(\delta^*)}{\tau} - \log\big(\sum_{s \in \mathcal{S}} \eta(s) \exp(\tau^{-1} f(s))\big).$$

As $\mathrm{KL}(\delta^* \, || \, \delta^*) = 0$, we have:

$$\mathcal{L}_\tau(\delta^*) = \max_{\delta \in \Delta_\mathcal{S}} \mathcal{L}_\tau(\delta) = \tau \log\big(\sum_{s \in \mathcal{S}} \eta(s) \exp(\tau^{-1} f(s))\big). \tag{19}$$

This result is often expressed in term of an inequality that says that the logsumexp is a majorant of the regularized expectation:

$$\forall \delta \in \Delta_\mathcal{S}, \tau \log\big(\sum_{s \in \mathcal{S}} \eta(s) \exp(\tau^{-1} f(s))\big) \geq \sum_{s \in \mathcal{S}} \delta(s) f(s) - \tau \mathrm{KL}(\delta \, || \, \eta). \tag{20}$$

## D.2 PROOF OF IMPROVEMENT.

We use the same notations as in the previous section, and our goal is to show that using the distribution $\delta^*$ instead of $\eta$ strictly increases the expected value of our function $f$:

$$\mathbb{E}_{s \sim \delta^*}[f(s)] > \mathbb{E}_{s \sim \eta}[f(s)],$$

unless $\delta^* = \eta$. This means that we can confidently replace $\eta$ by $\delta^*$ for the next iteration of the algorithm.

*Proof.* From Eq.(19), we have:

$$\tau \log \big( \sum_{s \in \mathcal{S}} \eta(s) \exp(\tau^{-1} f(s)) \big) = \mathcal{L}_\tau(\delta^*) = \mathbb{E}_{s \sim \delta^*}[f(s)] - \tau \mathrm{KL}(\delta^* \parallel \eta).$$

Using Jensen inequality we have:

$$\tau \log \big( \sum_{s \in \mathcal{S}} \eta(s) \exp(\tau^{-1} f(s)) \big) \geq \tau \sum_{s \in \mathcal{S}} \eta(s) \log \big( \exp(\tau^{-1} f(s)) \big),$$

$$= \tau \sum_{s \in \mathcal{S}} \eta(s) \tau^{-1} f(s),$$

$$= \sum_{s \in \mathcal{S}} \eta(s) f(s) = \mathbb{E}_{s \sim \eta}[f(s)].$$

This implies that:

$$\mathbb{E}_{s \sim \delta^*}[f(s)] - \tau \mathrm{KL}(\delta^* \parallel \eta) \geq \mathbb{E}_{s \sim \eta}[f(s)].$$

As $\tau \mathrm{KL}(\delta^* \parallel \eta)$ is strictly positive unless $\delta^* = \eta$, we conclude that we have strict improvement of the algorithm unless $\delta^* = \eta$ which means in this case that the method has converged. $\square$

## D.3 LINK BETWEEN IPO AND PMPO

In this section, we draw a parallel between the IPO and the PMPO losses. For a dataset of triplets $(x^i, y_a^i, y_r^i)_{i=1}^N$, the IPO loss is:

$$\mathcal{L}_{\mathrm{IPO}}(\theta) = \frac{1}{N} \sum_{i=1}^N \left[ -\log \big( \pi_\theta(y_a^i | x^i) \big) + \log \big( \pi_\theta(y_r^i | x^i) \big) + \beta \left( \log \left( \frac{\pi_\theta(y_a^i | x^i)}{\pi_{\mathrm{ref}}(y_a^i | x^i)} \right) - \log \left( \frac{\pi_\theta(y_r^i | x^i)}{\pi_{\mathrm{ref}}(y_r^i | x^i)} \right) \right)^2 \right].$$

The IPO loss is composed of two terms the policy optimisation term:

$$\mathcal{P}_{\mathrm{IPO}}(\theta) = \frac{1}{N} \sum_{i=1}^N -\log \big( \pi_\theta(y_a^i | x^i) \big) + \log \big( \pi_\theta(y_r^i | x^i) \big),$$

and the policy regularisation term:

$$\mathcal{R}_{\mathrm{IPO}}(\theta) = \frac{1}{N} \sum_{i=1}^N \left[ \left( \log \left( \frac{\pi_\theta(y_a^i | x^i)}{\pi_{\mathrm{ref}}(y_a^i | x^i)} \right) - \log \left( \frac{\pi_\theta(y_r^i | x^i)}{\pi_{\mathrm{ref}}(y_r^i | x^i)} \right) \right)^2 \right].$$

When $\alpha = \frac{1}{2}$, the PMPO loss can be written as:

$$\mathcal{L}_{\mathrm{PMPO}}(\theta) = \frac{1}{N} \sum_{i=1}^N \left[ -\frac{1}{2} \log \big( \pi_\theta(y_a^i | x^i) \big) + \frac{1}{2} \log \big( \pi_\theta(y_r^i | x^i) \big) + \beta \mathrm{KL}(\pi_{\mathrm{ref}}(.|x^i) || \pi_\theta(.|x^i)) \right].$$

The PMPO loss is also composed of two terms the policy optimisation term:

$$\mathcal{P}_{\mathrm{PMPO}}(\theta) = \frac{1}{N} \sum_{i=1}^N -\frac{1}{2} \log \big( \pi_\theta(y_a^i | x^i) \big) + \frac{1}{2} \log \big( \pi_\theta(y_r^i | x^i) \big),$$

and the policy regularisation term:

$$\mathcal{R}_{\text{PMPO}}(\theta) = \frac{1}{N} \sum_{i=1}^{N} \left[ \text{KL}(\pi_{\text{ref}}(.|x^i) || \pi_\theta(.|x^i)) \right].$$

Therefore, the policy optimisation terms of the IPO $\mathcal{P}_{\text{IPO}}(\theta)$ and PMPO $\mathcal{P}_{\text{PMPO}}(\theta)$ losses are identical at a constant factor. Now we are going to create a connection between the IPO and PMPO regularisation terms when $y_a^i$ and $y_r^i$ are sampled from $\pi_{\text{ref}}(.|x^i)$. The first thing to remark is that $\left( \log \left( \frac{\pi_\theta(y_a^i|x^i)}{\pi_{\text{ref}}(y_a^i|x^i)} \right) - \log \left( \frac{\pi_\theta(y_r^i|x^i)}{\pi_{\text{ref}}(y_r^i|x^i)} \right) \right)^2$ is an unbiased estimate of $E_{Y,Y' \sim \pi_{\text{ref}}(.|x^i)} \left[ \left( \log \left( \frac{\pi_\theta(Y|x^i)}{\pi_{\text{ref}}(Y|x^i)} \right) - \log \left( \frac{\pi_\theta(Y'|x^i)}{\pi_{\text{ref}}(Y'|x^i)} \right) \right)^2 \right]$. Then, we remind the reader that the variance of a random variable $X$ under distribution $\mu$ verifies:

$$\text{VAR}_{X \sim \mu}[X] = \frac{1}{2} \mathbb{E}_{X,X' \sim \mu}[(X - X')^2],$$

where $X$ and $X'$ are independent variables with distribution $\mu$. This means that $\left( \log \left( \frac{\pi_\theta(y_a^i|x^i)}{\pi_{\text{ref}}(y_a^i|x^i)} \right) - \log \left( \frac{\pi_\theta(y_r^i|x^i)}{\pi_{\text{ref}}(y_r^i|x^i)} \right) \right)^2$ is an unbiased estimate of $2\text{VAR}_{Y \sim \pi_{\text{ref}}(.|x^i)} \left[ \log \left( \frac{\pi_\theta(Y|x^i)}{\pi_{\text{ref}}(Y|x^i)} \right) \right]$.

Therefore the expectation (over $\pi_{\text{ref}}$) of the IPO regularization term is:

$$\mathbb{E}_{\pi_{\text{ref}}} \left[ \mathcal{R}_{\text{IPO}}(\theta) \right] = \frac{2}{N} \sum_{i=1}^{N} \text{VAR}_{Y \sim \pi_{\text{ref}}(.|x^i)} \left[ \log \left( \frac{\pi_\theta(Y|x^i)}{\pi_{\text{ref}}(Y|x^i)} \right) \right].$$

As $\text{VAR}_{X \sim \mu}[X] = \text{VAR}_{X \sim \mu}[-X]$, we also have:

$$\mathbb{E}_{\pi_{\text{ref}}} \left[ \mathcal{R}_{\text{IPO}}(\theta) \right] = \frac{2}{N} \sum_{i=1}^{N} \text{VAR}_{Y \sim \pi_{\text{ref}}(.|x^i)} \left[ \log \left( \frac{\pi_{\text{ref}}(Y|x^i)}{\pi_\theta(Y|x^i)} \right) \right].$$

This is in contrast with the regularization term of PMPO:

$$\mathcal{R}_{\text{PMPO}}(\theta) = \frac{1}{N} \sum_{i=1}^{N} \left[ \text{KL}(\pi_{\text{ref}}(.|x^i) || \pi_\theta(.|x^i)) \right] = \frac{1}{N} \sum_{i=1}^{N} \mathbb{E}_{Y \sim \pi_{\text{ref}}(.|x^i)} \left[ \log \left( \frac{\pi_{\text{ref}}(Y|x^i)}{\pi_\theta(Y|x^i)} \right) \right].$$

So the difference between PMPO and IPO is that PMPO will minimize the expectation of the log ratio between the reference policy and the online policy whereas IPO will minimize the variance of the same quantity.

# E BENCHMARKS

## E.1 CONTROL SUITE

The DeepMind Control Suite (Tunyasuvunakool et al., 2020) is a collection of benchmark tasks implemented in the MuJoCo simulator (Todorov et al., 2012). The suite includes a variety of embodiments of different complexity and action dimensionality. For each of these embodiments, there are multiple tasks implemented, each of which defines a single reward function. Example images for some of the domains are shown in Figure 6.

## E.2 RGB STACKING

The RGB Stacking benchmark (Lee et al., 2021) is a robotics task involving a Rethink Sawyer robot arm outfitted with a Robotiq 2F-85 gripper, as well as a basket containing a number of parameterized geometric shapes in red, green and blue color. See Figure 7 for an illustration. The goal of this task is to have the robot arrange the shapes into varying arrangements, such as stacking one on top of another or building a tower. The policy only provides proprioception information and the

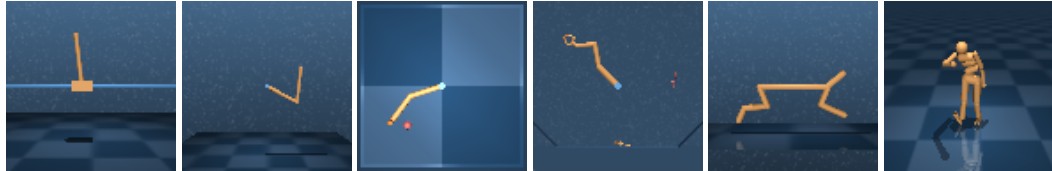

Figure 6: Example domains in the Control Suite. From left to right: Cartpole, Acrobot, Reacher, Manipulator, Cheetah, Humanoid

images from three cameras surrounding the basket; there is no explicit tracking of the objects' relative positions, or their identities. Specifically, the agent is provided the observations listed in Table 2. Thus the task's challenge lies in forcing the agent to learn a control policy directly from vision, and recognizing which objects are in the workspace, since their different geometric properties demand different manipulation strategies.

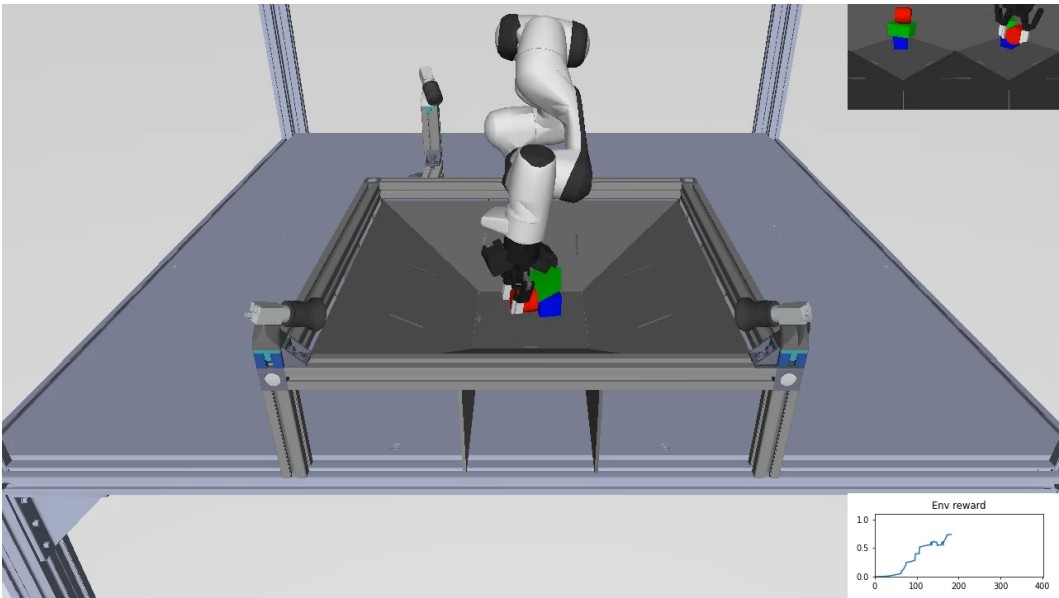

Figure 7: Illustration of the simulated RGB Stacking domain. Top right corner: goal image (left) and current observation (right) for the front left camera, both as provided to the agent. Bottom: reward trace for the dense triple-stacking objective.

The task is implemented in the MuJoCo physics simulator (Todorov et al., 2012). Using the ground truth positions of the objects' positions (which is not provided to the agent), several reward terms are calculated, including stacking two objects, stacking three objects, and building a pyramid, for each object permutation. Details of these rewards can be found in Springenberg et al. (2024).

## F    ADDITIONAL EXPERIMENTS

In this section we evaluate the performance of PMPO and DPO in high-dimensional control tasks from the DeepMind Control Suite (Figure 8). For each state, we sample four responses from a reference policy. We compare PMPO and DPO under two conditions:

1. Two Samples: Both algorithms utilize only two samples for learning; the best and worst responses (Accept[Best] & Reject[Worst]).

2. Four Samples: Both algorithms utilize all four samples (Accept&Reject). For PMPO, the top two responses are used as preferred generations and the bottom two as dispreferred

| Modality | Dimensions |
|---|---|
| Arm joint angles | 7 x 3 |
| Arm joint velocities | 7 x 3 |
| Arm joint torque | 7 x 3 |
| Gripper motor angle | 1 x 3 |
| Gripper motor velocity | 1 x 3 |
| Gripper grasp sensor | 1 x 3 |
| Cartesian tool center point pose | 7 x 3 |
| Cartesian wrist endpoint velocity | 6 x 3 |
| Cartesian wrist endpoint force | 3 x 3 |
| Cartesian wrist endpoint torque | 3 x 3 |
| Back left basket camera | 80 x 80 |
| Front left basket camera | 80 x 80 |
| Front right basket camera | 80 x 80 |

Table 2: Observations given to the agent in the RGB Stacking benchmark. Note that for all observations except the camera images, a history of 3 time steps is provided, resulting in the x3 dimensionalities.

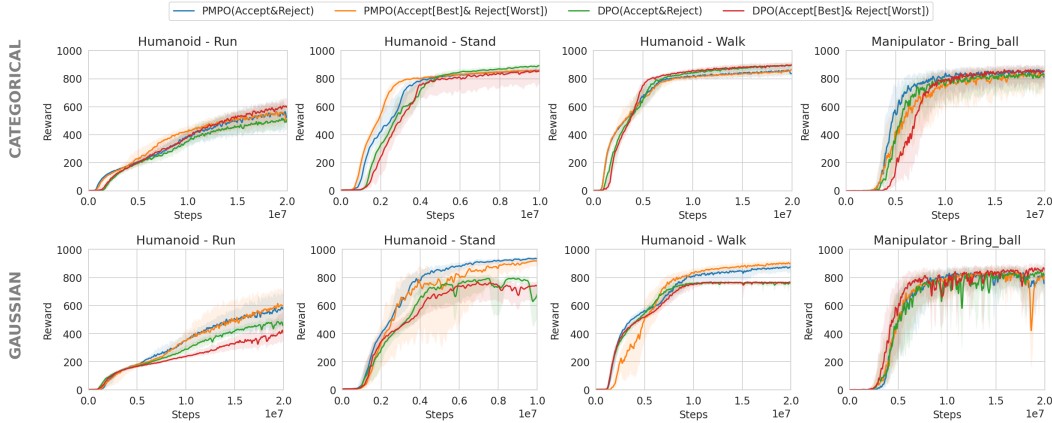

Figure 8: Comparison of PMPO and DPO for high-dimensional control tasks from the DeepMind Control Suite. We plot average reward over time of training (using 100 episodes for each evaluation). For each state we sample 4 responses from reference policy. We compare PMPO and DPO when both use only 2 samples for learning; the best and worst responses only (Accept[Best]&Reject[Worst]). We also compare PMPO and DPO when both use all 4 samples. For PMPO these would be top two as preferred generations and bottom two as dispreferred generations (Accept[Best]&Reject[Worst]). For DPO we create two sets of pairs, i.e., (best and worst) and (second best and second worst). Results show in general using two samples for learning and all 4 samples in learning yielded similar performance.

generations. For DPO, we create two sets of pairs for each prompt: (best and worst) and (second best and second worst).

Figure 8 indicates that using two samples for learning yields similar performance to using all four samples. Moreover, PMPO remains competitive with DPO when using pairs, while generally performing better when the policy is represented as Gaussian.

