# OpenReview forum: "Learning from negative feedback, or positive feedback or both"
_ICLR.cc/2025/Conference — ICLR 2025 Spotlight_

### Official Review · Reviewer_zTUL · 2024-10-31

**Soundness:** 2
**Presentation:** 3
**Contribution:** 2
**Rating:** 8
**Confidence:** 3

**Summary:**

The paper introduces a way to learn from unpaired preference data which is a constraint for other algorithms such as DPO. The method is motivated by expectation-maximization methods and results in an objective that weights positive and negative samples and applies cross entropy to positive samples and negative cross entropy to negative samples with KL regularization. They demonstrate their method on different benchmarks including bandit settings, DeepMind control suite, and language model alignment.

**Strengths:**

The paper addresses the problem of being able to use unpaired data and allowing for general preference distributions. In particular, they present an objective that is derived from maximizing an expectation that is motivated by existing methods. They perform experiments on multiple datasets and demonstrate that the theory is applicable through varying $\beta$ for only positive and only negative samples.

**Weaknesses:**

The main concern is whether the issue of unpaired preference data is a major problem and whether the experiments present a fair comparison between DPO and the proposed methods. In particular, if there are unpaired preference labels for a given state, it seems like a simple fix would be to pair each positive outcome with each negative outcome. Additionally, in the experiments, while the other baselines had access to all 4 samples, only DPO had access to 2. While the preferences should be paired, there is no restriction on having one pair of preferences per state. It would be more clear that the paper addresses an important issue if pairing samples from an unpaired dataset does have poor performance.

Furthermore, it would be more clear if it was mentioned that the reference model was updated as at the end of section 5.1, it is mentioned that there is a slowly changing reference. This varies from the original DPO which has a fixed reference.

There also seems to be a significant drop in performance with PMPO using both accepted and rejected responses in Figure 4. Furthermore, using all responses does not result in a higher peak than only using accepted responses which is concerning as it seems that using more data actually leads to worse performance. Additionally, there are seemingly incomplete or disrupted lines in Figure 5.

**Questions:**

1. How does using DPO with all (positive, negative) pairs from the 4 samples compare to PMPO?
2. Is there an issue with pairing positive and negative responses from unpaired datasets?
3. Why does using both accepted and rejected responses for language model alignment perform worse?
4. Can you explain Figure 5, in particular, the different cutoffs?

---

### Official Review · Reviewer_uKto · 2024-11-03

**Soundness:** 3
**Presentation:** 1
**Contribution:** 2
**Rating:** 5
**Confidence:** 3

**Summary:**

This paper presents a preference optimization method that utilizes unpaired examples and can learn from either positive or negative feedback, addressing limitations in existing approaches by that need paired data.

**Strengths:**

The proposed method allows for training with unpaired examples and accommodates scenarios where only one type of feedback—positive or negative—is available, making it widely applicable across different contexts.

**Weaknesses:**

* Personally, I find the presentation of this paper lacking. The main formulation in equation (10) is quite intuitive and provides a straightforward extension of the previous pairwise method to a more general setting. However, the derivations in Sections 3.1 and 3.2 are tedious and difficult to follow. I question the necessity of such extensive derivation from the expectation-maximization (EM) framework. It seems possible that the authors formulated the equation first and then sought a probabilistic framework to justify it. If this is the case, I strongly encourage the authors to present equation (10) prominently and follow it with a brief explanation of its connection to EM in a small subsection or appendix.

* I would particularly like to see a more rigorous comparison, using the Gemma-2B model, both methods could be trained on the pairwise UltraFeedBack / LMSYS-chat-1M dataset and then evaluated on the Alpaca Eval benchmark. If PMPO cannot match DPO, it will be important to delineate the limitations of PMPO and understand when it is appropriate to use this approach.

**Questions:**

See above

---

### Official Review · Reviewer_7waa · 2024-11-03

**Soundness:** 3
**Presentation:** 3
**Contribution:** 3
**Rating:** 8
**Confidence:** 3

**Summary:**

The paper proposes a preference optimization method that can utilize not only paired but also signal preferred or dis-preferred outcomes. The authors extend and improve on EM to tackle this problem and show empirical evidence favoring the proposed approach.

**Strengths:**

- Tackling the relevant and complex problem of incomplete data in preference optimization, for example, only having access to a negative examples
- Thorough and extensive related work making the contribution clear
- Objective is intuitive and makes sense probabilistically, especially through the use of the prior
- More flexible than methods like DPO and might apply to novel scenarios
- Extensive empirical evaluation on a variety of tasks from control, rl, to llm preference optimization

**Weaknesses:**

- Does introduce new hyperparameters that are potentially non-trivial to tune ($\alpha, \beta$)
- Title could be more specific. For example, something mentioning the capability to learn from dis-preferred examples. This could also help to attract readers interested in this particular problem. Currently, it seems only appealing to researchers interested in probabilistic inference.
- Does not improve over DPO, but might also be due to missing datasets well-suited for the setup

**Questions:**

- How to optimize $\alpha, \beta$ in practice?

---

### Official Review · Reviewer_DbEb · 2024-11-04

**Soundness:** 2
**Presentation:** 3
**Contribution:** 2
**Rating:** 8
**Confidence:** 3

**Summary:**

The paper proposes a preference learning objective (PMPO) that can utilize not just preference pairs but any combination of positive only or negative only samples. The objective is derived by defining an EM formulation for the expected success maximization objective of Eq 1 and defining the M step for both preferred and dispreferred samples. Experiments show that PMPO can operate with preference pairs as well as only preferred or only dispreferred samples.

**Strengths:**

1. The proposed PMPO and its derivation is, to my knowledge, novel. The objective is also easy to understand and implement, and the derivation has a clear probabilistic grounding in expectation maximization.
2. The paper is clearly written and tackles a relevant topic to the ICLR community.

**Weaknesses:**

While the method derivation is clear and well-motivated, the primary weakness in the work lies in the experiments.
1. The proposed method does not outperform DPO, the main baseline being compared to.
2. The experiments on bandit RL tasks focus on DPO as a baseline, without considering other methods used in these benchmarks.
3. The DPO baseline does not seem to use all the data given to PMPO; for instance, the end of Section 5.1 states that DPO uses "the best and worst action samples among the 4 sample archives", rather than all samples.
4. While PMPO can be adapted to a wider array of settings than DPO, for any given setting, it is not clear when and why one would choose to use PMPO over another method for that setting, e.g. PMPO on preferred only vs. SFT on preferred.
5. Assuming a sequence-level forward KL term, the proposed objective seems to simply amount to a weighted average of positive log prob terms for large enough $\beta$: namely, if $\mathcal{J} = \frac{1}{n} [\alpha \sum_{y \in D_a} \log \pi_\theta(y|x) - (1 - \alpha) \sum_{y \in D_r} \log \pi_\theta(y|x) - \beta \sum_{y \in D_a \cup D_r} \log \pi_{ref}(y|x) + \beta \sum_{y \in D_a \cup D_r} \log \pi_\theta(y|x)$,

then for $\beta > 1 - \alpha$, the objective is a sum of positive log probs only. Indeed, Figure 3 suggests that a large $\beta$ value is needed, which seems to suggest that PMPO works when it is close to supervised finetuning (with the only difference being a different non-negative weight for the preferred and dispreferred samples).

**Questions:**

1. Can the authors clarify if PMPO is indeed implemented as the expression in #5 above? If not, could the authors clarify the differences? If so, can the authors clarify its significance?
2. Can the authors clarify in what setting one would choose to use PMPO over other common baselines and why?
3. Why doe the authors focus on DPO as the baseline? Why not consider other common methods in RL benchmarks?

Other smaller comments:
1. In the first paragraph of experiments, shouldn't $\beta$ be $alpha$?
2. What is the value of $\beta$ used in Figure 2?
3. Why is the MPO baseline in Figure 2 denoted with a dashed horizontal line?

---

### Meta-Review · Area_Chair_TV5S · 2024-12-23

**Metareview:**

This paper extends the traditional preference optimization framework (e.g. DPO) to the case where preference data are not necessarily paired. The proposed algorithm, PMPO, is built on an EM formulation.

The reviewers generally agree that this is a significant problem to tackle, and it is solved in a novel and sound way. Indeed, there are practical situations where unpaired data are available, and it is only beneficial to have a framework that can take them into account. The authors also provide theoretical grounding for their approach.

For PMPO to work, two hyperparameters need to be tuned, however this is to some extent expected and they don’t seem to be too difficult to tune. A common concern of the reviewers has been that PMPO does not outperform DPO when pair data is available, however, the authors convincingly explain that the paper’s purpose is to provide a solution which uncouples learning from pairs of positive and negative feedback. Importantly, not only do the paper’s claims align with this view, but the authors have also promised to include a section on the applicability of PMPO and its limitations.

Overall, this paper promotes the state-of-the-art in learning from feedback by generalizing the learning setting, and thus it should be of interest to the ICLR community.

**Additional Comments On Reviewer Discussion:**

There has been extensive discussion and both the reviewers and authors were engaged. Various clarifications were discussed (including a change in the title of the paper), and overall this discussion improved the revised paper’s presentation.

Most importantly, the discussions focused on the weaknesses mentioned in the meta-review section, such as comparison with DPO. The reviewers are generally convinced by the motivation of the paper, since the authors explained “when it is appropriate to choose PMPO” and also agreed to add a section on the limitations of PMPO.

---

### Decision · Program_Chairs · 2025-01-22

Accept (Spotlight)